# Elucidating the Structure-Function Relationship of Solvent and Cross-Linker on Affinity-Based Release from Cyclodextrin Hydrogels

**DOI:** 10.3390/gels6010009

**Published:** 2020-03-22

**Authors:** Sean T. Zuckerman, Edgardo Rivera-Delgado, Rebecca M. Haley, Julius N. Korley, Horst A. von Recum

**Affiliations:** 1Affinity Therapeutics, LLC, 11000 Cedar Avenue, Suite 285, Cleveland, OH 44106, USA; stz@affinitytherapeutics.com (S.T.Z.); jnk@affinitytherapeutics.com (J.N.K.); 2Department of Biomedical Engineering, Case Western Reserve University, 10900 Euclid Avenue Cleveland, OH 44106, USA; edgardo.rivera@case.edu (E.R.-D.); rebecca.haley@case.edu (R.M.H.)

**Keywords:** affinity-based, drug delivery, polymer, cyclodextrin, minocycline, structure-function relationship

## Abstract

Minocycline (MNC) is a tetracycline antibiotic capable of associating with cyclodextrin (CD), and it is a frontline drug for many instances of implant infection. Due to its broad-spectrum activity and long half-life, MNC represents an ideal drug for localized delivery; however, classic polymer formulations, particularly hydrogels, result in biphasic release less suitable for sustained anti-microbial action. A polymer delivery system capable of sustained, steady drug delivery rates poses an attractive target to maximize the antimicrobial activity of MNC. Here, we formed insoluble hydrogels of polymerized CD (pCD) with a range of crosslinking densities, and then assessed loading, release, and antimicrobial activity of MNC. MNC loads between 5–12 wt % and releases from pCD hydrogels for >14 days. pCD loaded with MNC shows extended antimicrobial activity against *S. aureus* for >40 days and *E. coli* for >70 days. We evaluated a range of water/ethanol blends to test our hypothesis that solvent polarity will impact drug-CD association as a function of hydrogel swelling and crosslinking. Increased polymer crosslinking and decreased solvent polarity both reduced MNC loading, but solvent polarity showed a dramatic reduction independent of hydrogel swelling. Due to its high solubility and excellent delivery profile, MNC represents a unique drug to probe the structure-function relationship between drug, affinity group, and polymer crosslinking ratio.

## 1. Introduction

Minocycline (MNC) is a second generation tetracycline that has longer half-life than first generation combined with better oral availability and fewer resistance concerns [1]. MNC acts by blocking aminoacyl-tRNA association with the bacterial ribosome, thereby inhibiting protein synthesis. MNC has a broad spectrum of activity, including coverage of common Gram positive bacteria, such as *Staphylococcus aureus* (*S. aureus*), and coagulase-negative *Staphylococci* (CoNS), such as *S. epidermidis*. Most importantly, MNC has activity against both community-acquired and nosocomial methicillin-resistant *S. aureus* (MRSA) [1,2,3]. In addition to its broad-spectrum activity, MNC also has several properties that make this drug ideal for localized delivery, including long half-life and time-dependent killing. The ratio of MNC’s area-under-the-curve to minimum inhibitory concentration (AUC/MIC) is the best predictor of antimicrobial activity [1]. A polymer delivery system capable of sustained, steady drug delivery rates poses an attractive target to maximize the antimicrobial activity of MNC. 

Numerous groups have delivered MNC with other antibiotics most notably rifampicin, including several devices currently on the market [4,5,6,7,8]. Rifampicin and MNC provide robust Gram-positive coverage, as demonstrated in the reduction in brain shunt infections. However, conventional delivery polymers rely solely on diffusion as the rate limiting step in release, resulting in bi-phasic, nonlinear delivery. When these conventional polymers are formulated into micron-thin coatings, such as coatings on drug eluting stents or coatings on catheters, release rates are quick as the drug’s diffusional path is very short. One traditionally used route to alter diffusion in such a scenario is to alter the crosslinking of the coating material, which also impacts the polymer’s mechanical properties.

Our group has previously synthesized and characterized polymers comprised solely of cyclodextrin (CD) and a crosslinker that uses the molecular association between drug and the hydrophobic interior of the CD to further control the rate of drug release. Release from this polymerized CD (pCD) is dictated by rate of association/dissociation of drug to the CD pocket; once the drug is dissociated, it is free to diffuse out of the polymer or associate with another CD pocket. This type of release mechanism is called affinity-based release [9] and is capable of loading high amounts of the drug, as well sustaining release beyond time frames achievable with conventional (diffusion only) delivery polymers. This effect is most prominently noticed in hydrogels where, due to high water content, drug mobility is particularly high and release is rapid. However, we have previously shown that, when molecular dissociation (affinity) is the rate limiting step, delivery even from hydrogels is slow and sustained [10]. In addition, the affinity interaction between drug and CD allows for the polymer to be filled post-implantation, even when the polymer is coated with a biofilm [11].

Since MNC is the most lipophilic member of the tetracycline family, we hypothesized that MNC would associate strongly with CD pockets and provide sustained release from pCD, prolonging antimicrobial activity. MNC also poses a unique molecule to explore the mechanism of affinity-based release since MNC is equally soluble in water and ethanol up to the concentration used in this work (20 mg/mL). Blending ethanol and water in various ratios, from 50:50 to 90:10, enabled us to explore the impact of solvent polarity and hydrogel swelling upon drug loading. Blending ethanol into water shifts solvent polarity towards less polar, which alters the binding equilibrium between drug and CD pocket and also alters hydrogel swelling, enabling us to begin elucidating the structure-function relationship for pCD.

## 2. Results

### 2.1. Equilibrium Water Content

pCD synthesis and characterization has been previously described [11,12,13,14]. Briefly, epichlorohydrin crosslinked CD prepolymer is dissolved in anhydrous dimethylformamide (25 % *w*/*v*) and crosslinked with varying ratios of hexamethylene diisocyanate. This work focused on the antimicrobial activity of MNC released from pCD, determining the effect of increasing cross link density upon affinity-based release and the impact of solvent polarity on loading and swelling. pCD swelling in water decreased with increasing crosslink density for pCD-α, -β, and -γ (Figure 1). Two-way ANOVA revealed that both CD type (α, β, or γ) and cross linking were significantly different (*p* < 0.05), with the exception of pCD at 1:0.32 and 1:0.64, which bordered on significance (*p* = 0.053). pCD-β at the highest crosslinking density (1:0.64; 144.8 ± 12.1%) swelled significantly more than pCD-α (80.1 ± 5.9%) or -γ (58.9 ± 3.2%) (*p* < 0.001), and, overall, pCD-β showed the smallest decrease in swelling with increasing crosslinking (Figure 1). 

### 2.2. Rheological Studies

The storage modulus (G’) and loss modulus (G’’) also vary by type of pCD. Figure 2 shows that both G’ and G’’ increase for all three types of pCD polymers as a function of angular frequency (ω) at constant shear (τ = 5 Pa). Storage modulus is more than an order of magnitude greater than the loss modulus, indicating that pCD polymers are more elastic than viscous. For G’, the trend is pCD-γ >> pCD-α > pCD-β. G’’ shows a similar trend by order pCD-γ > pCD-α > pCD-β. pCD-γ had the highest storage and loss, which is in line with γ-CD having the largest ring with the most flexibility. γ-CD also has the ability to crosslink in more places with less steric hinderance. 

MNC was chosen for the clear trend in binding affinity shown by Autodock binding simulation (Figure 3), where MNC binds to αCD (−5.1 kcal/mol) < βCD (−6.0 kcal/mol) < γCD (−7.4 kcal/mol). The Autodock simulations do not account for the potential impact of cross linking on pocket size, accessibility, or binding multiple pockets. 

MNC loading to pCD hydrogel disks was analyzed by extensive leaching into water followed by UV-Vis quantification. Increasing cross linking resulted in significantly decreased MNC loading as a weight percent for 1:0.16, 1:0.32 and 1:0.64 (Figure 4). There was no statistical difference between loading for 1:0.08 and 1:0.16 (*p* = 0.1). In addition, two-way ANOVA showed no difference between pCD-α and pCD-β loading (*p* = 0.95), but both were significantly different from pCD-γ (*p* <0.001 and =0.001, respectively).

Cumulative MNC release normalized to polymer weight varied by CD type and cross linking. All crosslink ratios showed varying levels of initial burst release followed by steady, affinity-based release. pCD-α at 1:0.08 released more MNC than pCD−β or –γ (Figure 5). MNC release from pCD–β increased with increasing crosslink density and approached the amount and rate from pCD–α by 1:0.32. More MNC released from pCD–β at 1:0.64 than pCD−α or –γ. 

MNC-loaded pCD at 1:0.32 showed extended in vitro bioactivity in a zone of inhibition assay against both Gram-positive *S. aureus* and Gram–negative *E. coli* bacteria (Figure 6). MNC released from pCD cleared a zone on *S. aureus* lawns > 4 mm for 40 days. MNC activity against Gram-negative *E. coli* persisted for 70 d. 

We chose pCD-β to investigate the impact of swelling and solvent polarity on loading because pCD-β showed the smallest decrease in MNC loading (Figure 4) and swelling (Figure 1), with increasing crosslink density from 1:0.08 to 1:0.64. MNC is soluble in both ethanol and water, but pCD does not swell in ethanol. pCD swelling in blends of ethanol:water was first determined at range of 50:50 to 90:10 (Figure 7A). pCD with 1:0.08 crosslinking demonstrated increased swelling at 50:50 ratio of ethanol:water (*p* < 0.015) and comparable swelling to pure water at 65:35 ethanol:water (257 ± 37 % pure water vs. 229 ± 10% in 65 : 35 ethanol : water; *p* < 0.004 for 50:50 to 65:35). The swelling showed a significant decrease at 90:10 ethanol: water (46 ± 4%; *p* < 0.0001 for all other ethanol:water blends)). pCD at 1:0.64 crosslinking showed a modest, but significant, overall decrease in swelling from 145 ± 12% in pure water to 113 ± 10% in 90 :10 ethanol:water (*p* < 0.0005). 90:10 ethanol:water was significantly lower than pure water (*p* < 0.007). 50:50 ethanol:water swelled slightly more than pure water but bordered on significance (*p* = 0.053). The 1:0.08 pCD showed a significant decrease in MNC loading going from 11.3 ± 3.8 wt % in pure water to 4.0 ± 0.2% in 50:50 water:ethanol and further down to 2.2 ± 0.9% in 90:10 ethanol:water (*p* < 0.05), but there was no significant difference in MNC loading among 50:50, 65:35, nor 90:10 (*p* = 1). MNC loading in 1:0.64 pCD-β was significantly higher, at 50:50, compared to pure water and 90:10 ethanol:water. In addition, the 90:10 ethanol:water was lower than pure water (*p* < 0.0005). 

## 3. Discussion

MNC is a well-known broad-spectrum antibiotic and is the most hydrophobic tetracycline family member making it an ideal candidate for affinity-based delivery from pCD. Previous work using pCD focused on antibiotics with Gram-positive coverage like rifampicin and vancomycin [12,13,15]. In this paper, we show that MNC delivered from pCD was capable of killing Gram-positive *S. aureus* for at least 40 days and Gram-negative *E. coli* for at least 70 d (Figure 6). This approach represents a promising strategy for delivering antibiotics in light of the fact that our group has also previously shown that the affinity interactions present in pCD are capable of driving MNC filling into empty pCD after implantation, even when the pCD is coated with a mature biofilm [11]. Sustained delivery of MNC from pCD is ideally suited for biofilm penetration, which is critical in the case of an established biofilm on the surface of an implant to ensure complete eradication and avoid resistance. 

To ensure the appropriate amounts of MNC are loaded onto the infected biomedical device, we investigated how different formulation parameters alter the loading of MNC at physiologic pH. pCD has previously been shown to swell extensively in water and organic solvents, such as DMF [13,16]. This previous work focused on using the swelling to facilitate equilibrium drug loading. We hypothesize that the swelling also has an effect on the 3-dimensional shape of CD, as well as possible caging by the crosslinker that will both have an impact on drug loading [17]. To investigate this hypothesis, a series of ethanol:water blends from 50:50 to 90:10 were prepared to achieve a wide range of swelling across the highest (1:0.64) and lowest (1:0.08) crosslinking ratios tested in this work (Figure 7). The lowest crosslinking of all pCD types (1:0.08) exhibited a broad range of swelling from >250% to roughly 50% across the ranges of ethanol:water. In particular, a significant decrease between 65:35 and 90:10 ethanol:water ratios was observed. A smaller drop was seen for pCD at 1:0.64 from 50:50 to 90:10 ethanol:water. The swelling for pCD 1:0.64 was significant (*p* < 0.05) but is relatively consistent across ethanol:water blends when compared to swelling seen for pCD at 1:0.08 (~50–350%). Pure ethanol was not tested for MNC loading as it does not appreciably swell pCD. Thus we believe pCD crosslinked at 1:0.08 shows the impact of swelling (~50–350%) upon MNC binding affinity for pCD, while pCD crosslinked at 1:0.64 shows the impact of solvent polarity on MNC binding to pCD. 

We hypothesized that shifting the polarity index from water (polarity index 9) toward ethanol (polarity index 5.2) would gradually decrease MNC loading. MNC solubility did not vary across the blends of ethanol:water, but the presence of ethanol significantly decreased MNC loading to pCD for both 1:0.08 and 1:0.64 (Figure 7). However, Figure 7B shows that polarity plays a dominant role in determining drug binding to pCD compared to the impact of swelling upon CD shape since MNC loading drops at 50% ethanol and is relatively constant up to 90% ethanol. Blends of water and ethanol are commonly used to tweak solvent polarity when studying the association of free compounds with soluble CD [18,19,20]. Ethanol is capable of weakly associating with the CD pocket; however, Autodock simulations calculated significantly lower binding energies for ethanol than MNC (−1.6, −1.7 and −1.5 kcal/mol for pCD-α, -β and –γ, respectively), so any ethanol occupying the pocket would be displaced by minocycline. We hypothesize that ethanol is tilting the thermodynamic equilibrium away from MNC-CD association by decreasing the energetic favorability of desolvating MNC by associating into the CD pocket [21,22].

While attention has focused on binding equilibrium in solution, the crosslinked form of pCD, or any other host-guest-based affinity polymer, could exhibit binding energies that differ from those in solution. To illustrate this point, we loaded MNC dissolved in pure ethanol into lightly epichlorohydrin-crosslinked CD prepolymer, which is insoluble in ethanol, and observed very high loading – 34 wt % for α-, β- and γ-CD, respectively. Even light crosslinking significantly reduces the ability of CD to bind drug as shown by the ~12 wt % MNC loading into 1:0.08 pCD, but this loading reduction is smaller as crosslinking increases (Figure 4). But this decreased loading with increased crosslinking may also impact the burst release (Figure 5). With affinity-based release, the burst is a two-pronged phenomenon. Part of the burst release is drug adsorbed to the polymer and/or crosslinker (i.e., not part of an inclusion complex with CD), which also happens with non-affinity polymers; the other part is drug dissociating from a CD pocket and diffusing away from the polymer rather than re-forming an inclusion complex with another CD pocket. When the pCD polymer is initially hydrated, the ratio of drug:CD pockets is relatively even, which means fewer open CD rings are available for forming a new inclusion complex resulting in drug diffusing away and releasing. As this ratio tilts towards an excess of CD pockets, the rate of drug release slows as drugs are likely to form multiple inclusions prior to diffusing away from the polymer [23]. So, in this work, as polymer crosslinking increases and MNC loading decreases, the ratio of MNC:CD pockets favors an excess of CD pockets more quickly at higher crosslinking ratios evidenced by the decreased burst release seen from Figure 5B–D. The decrease is most pronounced for pCD-γ, which we hypothesize is due to reduced ring flex resulting from more crosslinking since CD-γ is the largest ring and has the most ring flex of the three main CD types. The crosslinked nature of pCD poses unique challenges to assessing drug-CD complex formation relative to soluble CD-drug inclusion measurements. 

Affinity-based release has the potential to load high amounts of water and organic solvent-soluble drugs. Understanding the structure-function relationship between CD type, cross linking, and drug is necessary to predict a priori which pCD polymer will deliver a given drug for the desired profile. The work presented herein is the first step to understanding this relationship for pCD and MNC. In addition, understanding the impact of crosslinker, swelling and solvent polarity will enable more tailored and efficient drug loading. Currently, we employ equilibrium drug loading that relies on high concentrations of drug in solution to achieve high drug loading in the pCD. A fuller understanding of the structure/function relationship of drug to pCD will allow thermodynamically driven loading, whereby the drug is more favorably associated with a CD pocket than being freely solvated in solution. We hypothesize that the loading solvent can be tailored such that the only requirement is a slight molar excess of drug relative to pCD. This knowledge will also potentially enable modeling and predicting in vivo filling and re-filling [11]. To date no conventional polymer has been shown to fill drug into empty polymer. Our group has previously shown that pCD can fill a drug into empty polymer even when coated with a biofilm [11]. 

Understanding the structure/function relationship’s impact on drug loading into our pCD polymer will help optimize the clinical use of this polymer, where the crosslinking chemistry may change to tailor polymer degradation, drug release rates and host response to address the demands of the specific clinical application. Degradation and host response to implanted materials can significantly alter release behavior. To date, no excessive host response has been found in rodents [15] or pigs [24] out to 30 days and preliminary data in pigs at 60 d shows no change from 30 d (unpublished data). The degradation products of pCD are glucose and do not elicit acidification or inflammatory response. 

## 4. Conclusions

Minocycline is capable of loading between 5–12 wt % into pCD and releasing for ≥ 2 weeks. MNC released from pCD is capable of killing Gram-positive bacteria for at least 40 d and Gram-negative bacteria for at least 72 d in vitro. This work also showed that, similar to solution-based drug-CD association, solvent polarity is critical to drug associating with CD pockets but is complicated by hydrogel swelling and crosslinking. Future work will focus on adapting solution-based binding energy techniques to insoluble pCD to define the effect of solvent on drug association and dissociation.

## 5. Materials and Methods

### 5.1. Materials

Epichlorohydrin-crosslinked α–, β–, and γ–Cyclodextrin prepolymer (CycloLab R&D) were used as received. The following items were purchased from Fisher (Waltham, MA, USA) at reagent grade and used as received: acetone; 1,6 diisocyanatohexane (HDI); dimethylformamide (DMF); hexanes; light paraffin oil; methylsulfoxide (DMSO); potassium hydroxide (KOH). MNC was purchased from Research Products International (Mt. Prospect, IL, USA).

### 5.2. Polymerized CD (pCD) Synthesis

CD prepolymer was dried overnight in a vacuum oven at 60 °C, dissolved in DMF (25 wt %), and mixed with HDI at various ratios of isocyanate:glucose residue (1:0.08, 1:0.16, 1:0.32, and 1:0.64). CD was allowed to crosslink for 4 days at room temperature in a 6 cm Teflon dish covered tightly with Parafilm. Disks (8 mm) were punched with a tissue biopsy punch and washed with excess DMF, DMF/water, and water to remove unreacted products and solvent. Disks were dried two days at room temperature prior to use.

### 5.3. Equilibrium Water Content

The equilibrium water content was evaluated by immersing the known weight (W_1_) of 8 mm diameter dried polymer hydrogel disks in water, DMF, and DMSO. After 24 h, samples were removed and blotted lightly with filter paper to remove excess water or DMF or DMSO and weighed again (W_2_). Equilibrium swelling was calculated as (W_2_ − W_1_)/W_1_. These studies were performed in triplicate (*n* = 3), with the average values reported.

### 5.4. Rheological Studies

Rheological characterization of 1:0.32 crosslinked pCD hydrogel disks was done using a controlled stress rheometer (TA instruments AR2000(ex) rheometer) equipped with stainless steel parallel plate geometry with 12 mm diameter and a solvent trap. The rheological characteristics of the water swollen disks were monitored by oscillatory frequency sweep experiment at 25 °C performed at a constant stress of 5 Pa (located in the range of linear viscoelasticity). Frequencies varied from 0.1–85 Hz and were plotted as angular frequency (0.6–535 radians/s). All the rheological measurements were performed in triplicate and the presented results are the average of those experiments.

### 5.5. Autodock Binding Simulation

PyRx software was used to convert minocycline’s structure data file from Pubchem into PDBQT format [25]. Autodock Vina algorithm calculated the binding energy for α-CD, β-CD, and γ-CD [26].

### 5.6. Minocycline Loading

pCD hydrogel disks were loaded with MNC (2 wt % in water) for three days at room temperature protected from light. MNC-loaded disks were blotted on a Kimwipe, dipped in Milli-Q water to remove non-bound MNC on the disk surface, blotted again, and dried at room temperature while covered from light. MNC was then leached into methanol and quantified via UV-vis spectrophotometry. Loading percent was calculated as: % drug loading = 100 × (W_leached drug_)/(W_leached drug_ + W_polymer_), where W_leached drug_ and W_polymer_ are weight of the drug leached from the loaded disk and weight of the polymer sample, respectively.

### 5.7. Minocycline Release

MNC-loaded pCD hydrogel disks were placed in 10 mL phosphate buffered saline (PBS, pH 7.4) at 37 °C with 100 RPM shaking. Samples were covered from light, and 0.5 ml removed at 1, 2, 4, 8, 24 and every 24 h thereafter. Sample aliquots were replaced with fresh PBS. To account for MNC degradation over time [27], a 5 mg/mL stock solution of MNC was also incubated with release samples and an aliquot taken at each time point to create a time-specific standard curve. Release aliquots were analyzed via UV-Vis spectrophotometry at 370 nm.

### 5.8. Zone of Inhibition Assay

A fresh bacterial lawn was created by spreading 70 μl *S. aureus* grown overnight in soy trypticase broth onto a sterile soybean-casein digest medium (BD BBL Trypticase) agar plate. MNC-loaded pCD hydrogel disks were placed directly on the lawn and allowed to incubate for 24 h. Calipers were used to measure the zone of inhibition prior to transferring the disk to a fresh lawn. *E. coli* lawns were created in a similar manner on lysogeny broth (LB) agar plates. 

### 5.9. Statistical Analysis

Two-way analysis of variance (ANOVA) with post-hoc Bonferonni correction was used with *p* ≤ 0.05 considered statistically significant.

## Figures and Tables

**Figure 1 gels-06-00009-f001:**
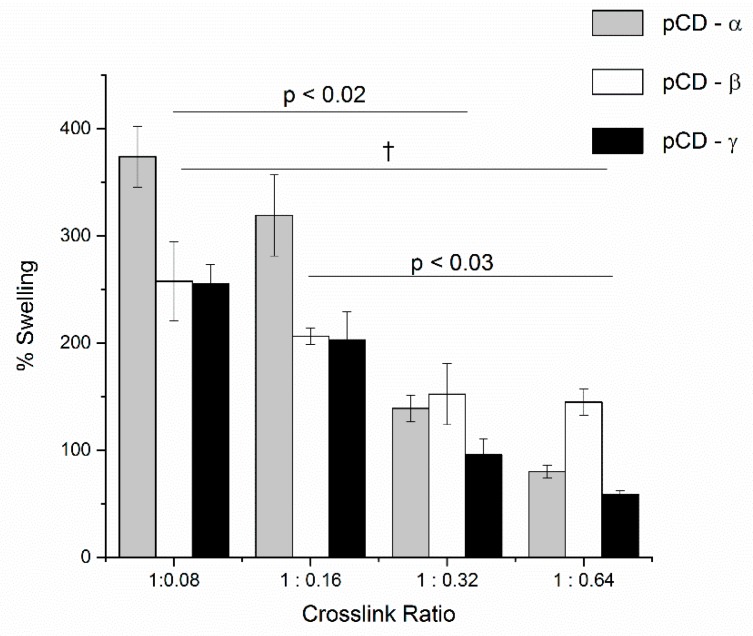
Increasing ratios of polymerized cyclodextrin (pCD) crosslink density decreases water swelling. pCD hydrogel disks were weighed dry (W_1_) placed in water at 37 °C for 24 h, blotted, and weighed (W_2_). Swelling was calculated according to (W_2_ − W_1_)/W_1_. Two-way analysis of variance (ANOVA) revealed the following comparisons as not significant: 1:0.08 vs 1:0.16 (*p* = 0.5); 1:0.16 vs 1:0.32 (*p* = 0.08); and 1:0.32 vs 1:0.64 (*p* = 0.8). † *p* < 0.01. Data shown is average ± std deviation for *n* = 3.

**Figure 2 gels-06-00009-f002:**
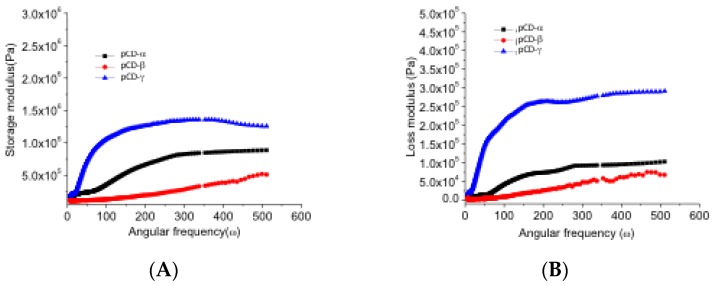
Frequency dependence of dynamic storage modulus (G’) (**A**), and loss modulus (G’’) (**B**) increased for all three types of pCD crosslinked at 1:0.32 according to the trend pCD-γ >> pCD-α > pCD-β. Viscoelastic properties of water swollen pCD hydrogel disks were measured using an oscillatory frequency sweep experiment at 25 °C, performed at a constant stress of 5 Pa (located in the range of linear viscoelasticity). Frequencies varied from 0.1–85 Hz plotted as angular frequency (0.6–535 radians/s). All measurements were performed in triplicate, and the presented results are the average of those experiments.

**Figure 3 gels-06-00009-f003:**
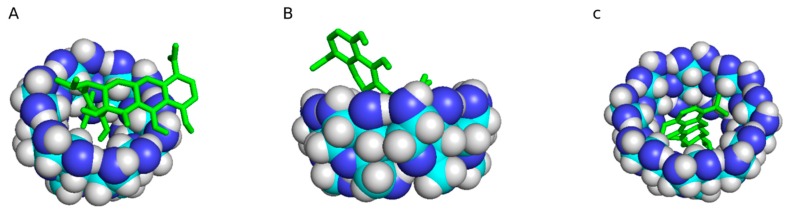
Autodock binding simulations showed a clear increasing trend in binding energies. Minocycline binds to (**A**) αCD (−5.1 kcal/mol); (**B**) < βCD (−6.0 kcal/mol); and (**C**) < γCD (−7.4 kcal/mol). The Autodock simulations do not account for the potential impact of cross linking on pocket size, accessibility, or binding multiple pockets, which were explored empirically.

**Figure 4 gels-06-00009-f004:**
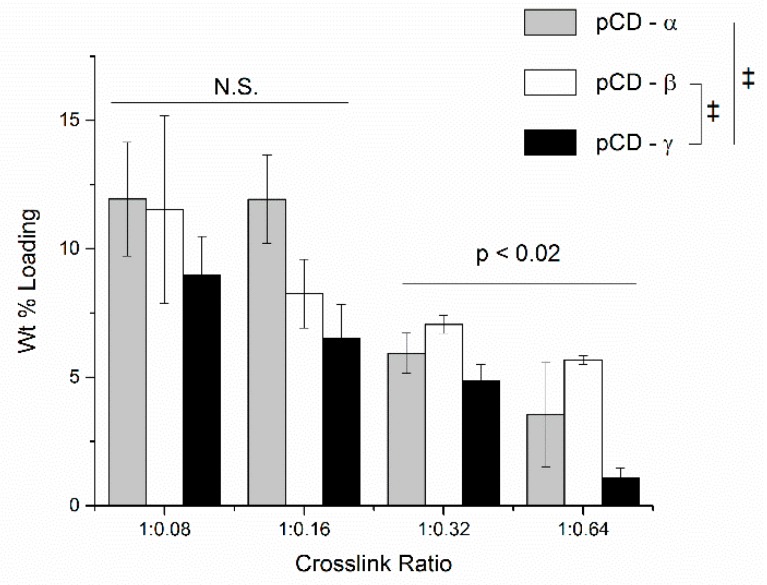
Minocycline loading decreased with increasing pCD crosslinking ratio. Briefly, minocycline (MNC) (20 mg/mL) was loaded into polymerized cyclodextrin (pCD) for three days prior to washing, drying, and leaching extensively into water. MNC was quantified by UV/Vis. Although 1:0.08 was not significantly different than 1:0.16, 1:0.16 showed more loading than 1:0.32, which showed more loading than 1:0.64. MNC loading to pCD-α was similar to pCD-β (*p* = 0.95) but was significantly different than pCD-γ (*p* < 0.001 and = 0.001, respectively). Data shown is average ± std deviation for *n* = 3. N.S. Not significant; ‡ p ≤ 0.001.

**Figure 5 gels-06-00009-f005:**
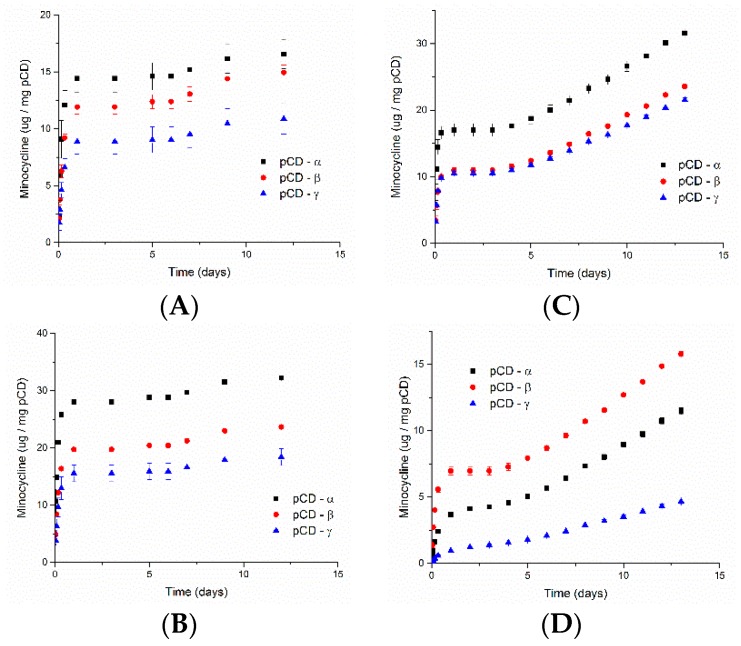
Cumulative minocycline (MNC) release normalized to pCD weight. The trend for release changed with increasing pCD crosslink ratio. At 1:0.08 (**A**) MNC released pCD-α > pCD−β or −γ as crosslinking increased from 1:0.16 (**B**) MNC released from pCD–β approached that from pCD-α by 1:0.32 (**C**) and at 1:0.64 (**D**) pCD–β > pCD-α or –γ. MNC was loaded three days at room temperature prior to releasing into phosphate buffered saline (PBS) at 37 °C. An aliquot was removed at each time point and replaced with fresh PBS to maintain constant volume. Data shown is average ± std deviation for *n* = 3.

**Figure 6 gels-06-00009-f006:**
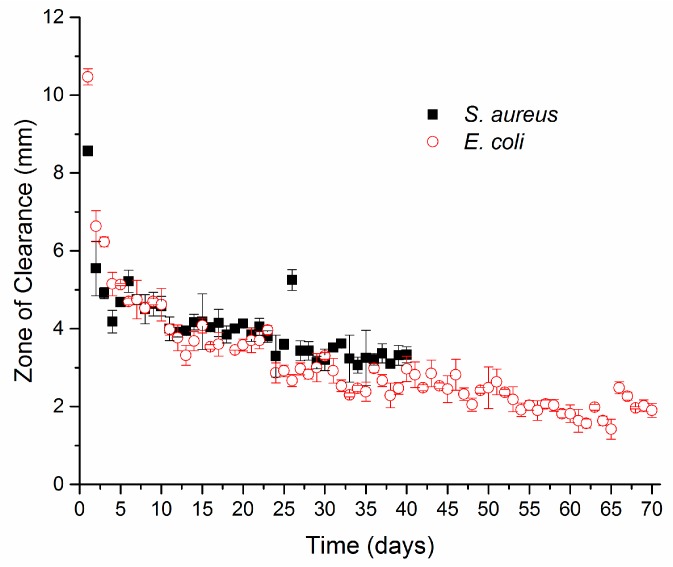
In vitro bioactivity of MNC released from pCD-β at 1:0.32 showed extended clearing of both Gram-positive *S. aureus* for > 40 d and Gram-negative *E. coli* for > 70 d. Each day, MNC-loaded pCD hydrogel disks were placed on a fresh bacteria lawn, incubated for 24 h, and had the zone of clearance measured with calipers prior to repeating the process. All samples were replicated three times with average ± std deviation (mm) shown.

**Figure 7 gels-06-00009-f007:**
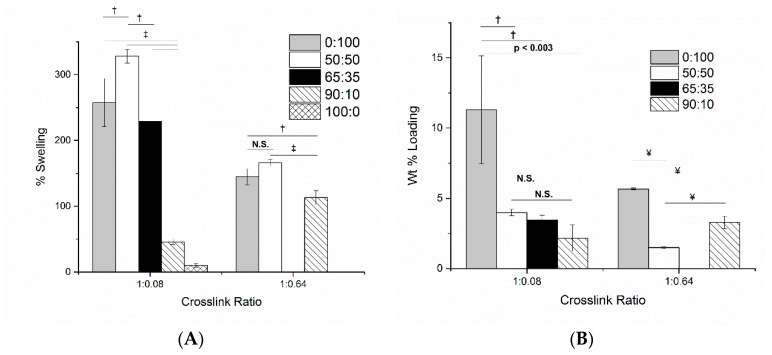
Solvent polarity has significant effects upon both pCD swelling (**A**) and minocycline (MNC) loading (**B**). pCD-β was chosen because it showed the least change in MNC loading and swelling across different crosslinking ratios. N.S. not significant (*p* = 0.9); † *p* < 0.01; ‡ *p* < 0.001; ¥ *p* < 0.0001. Data shown is average ± std deviation for *n* = 3.

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
