# Peer review of "Elucidating the Structure-Function Relationship of Solvent and Cross-Linker on Affinity-Based Release from Cyclodextrin Hydrogels"

_gels, 2020, doi:10.3390/gels6010009_

Round 1

Reviewer 1 Report

This is an interesting work for the use of commercially available cyclodextrin hydrogels for the controlled and sustained delivery of minocycline. I think it can be published in GELS after minor revisions.

The authors should describe better how they obtained the drug loading at 2.6. They mention the procedure of loading. The amount of loaded drug is the total amount of drug added minus the one released after blotted and dipped in water? How did they calculate this? The authors should mention the ratio of cyclodextrin molecules to drug loaded, like they mention in conclusions, as a parameter to be investigated. This can be made here. Maybe this will explain why at Figure 7B the amount of drug loaded increases slightly with increasing amount of alcohol. Which is the highest amount of drug that can be loaded? The authors should discuss more the results of Figure 5. For the hydrogels with lower degree of crosslinking the profile of release is a burst-type, while for 1:0.64 is a mixture of burst type after a few hours at the beginning and sustained release later.

Author Response

Thank you for the positive, and helpful review of our manuscript.  We hope we were able to address any remaining concerns:

>>The authors should describe better how they obtained the drug loading at 2.6...

We agree that the description was incomplete and have included a more detailed description.

>>The authors should mention the ratio of cyclodextrin molecules to drug loaded...

Agreed.  We have included a fairly lengthy discussion of this and how it may explain the alcohol results.

>>The authors should discuss more the results of Figure 5...

Agreed, and consistent with reviewer 3 as well.  We have added further discussion here too.

Reviewer 2 Report

The authors studied the release behaviours of MNC from cyclodextrin, in which the release of MNC is affinity-based. To elucidate the affinity interaction between MNC and CD, the authors systematically studied the effects of solvent and cross linker on the CD hydrogel which ultimately affect its affinity to MNC. 

I would like to recommend publication of the manuscript after the following concerns are addressed:

(1) The quality of Figure 2 should be improved.

(2) In Figure 5, the initial burst release of MNC is still high (I would like to recommend that the author show the initial burst release in a scale of hours, can be as an inset in the main figures). Is there some ways to control the burst release, since burst release is always not desirable in many situations. The ratios in Figure 5C & Figure 5D showed positive signs. After certain degree of burst release, the release of MNC in the following days exhibit a zero-order release manner which is quite useful for treatment purpose. It is suggested that the authors can discuss more about Figure 5. 

Author Response

Thank you for your positive review and helpful comments.

>> (1) The quality of Figure 2 should be improved. 

Done. This seems to have been a compression/decompression issue.  Please let me know if it hasn't been successfully resolved.

>>(2) In Figure 5, the initial burst release of MNC is still high (I would like to recommend that the author show the initial burst release in a scale of hours, can be as an inset in the main figures).

The burst release is near instantaneous. In our work some burst is desirable to rapidly bring a "patient" to some effective threshold level of drug (.e.g an antibiotic).  In past work we have shown that this burst is heavily controlled by excess, non-complexed drug and can be reduced by either loading less drug into the device in the first place, or having additional washing steps to further remove non-complexed drug.  Some of this discussion is in our increased discussion around Figure 5.

Reviewer 3 Report

The authors present the use of cyclodextrin (CD) hydrogels with different degrees of crosslinking and CD cavity sizes for release of minocycline (MNC) encapsulated within the cyclodextrin cavities. The release efficacies at different water-ethanol ratios are evaluated. Although the study was competently performed, the results presented are a routine follow-up work on previously-reported CD hydrogels from the same group for drug release and do not show any significant advancements in the field. Nonetheless, this work should be considered for publication after addressing some minor points:

Abstract line 24: please define pCD in the abstract. Intro line 33 (pg 1): insert 'that' between 'tetracycline' and 'has' Intro line 35 (pg 1): delete 'by'. Pg 1 line 36-37: 'broad spectrum....(CoNS)'. Please re-write- I am not sure what the authors mean in this phrase. Pg 4 section 3: the authors should include the structure of the cyclodextrin hydrogels to make it clearer to the readers what the polymers actually are. The stnthesis should also be briefly described instead of simply referring to previous publications- this will make it easier for readers to comprehend the work without having to follow the links. I would like to see evidence presented herein that MNC is encapsulated within the cyclodextrin cavities instead of simply interacting with the polymer without formation of host-guest complex.

Author Response

Thank you for your careful review and helpful comments.

>> Abstract line 24 (and other, similar edits)   Thanks for catching these.  We made the changes suggested.     >>The synthesis should also be briefly described instead of simply referring to previous publications- this will make it easier for readers to comprehend the work without having to follow the links.   Agreed.  We have included a more detailed description of the synthesis.     >> I would like to see evidence presented herein that MNC is encapsulated within the cyclodextrin cavities instead of simply interacting with the polymer without formation of host-guest complex.   While not specifically studied in this work, we point to past work including FTIR, DSC, and XRD confirming complex formation.